# Simulation of phased alerting of community first responders for cardiac arrest

Pieter L. van den Berg[1]*, Shane G. Henderson[2], Hemeng Li[2], Bridget Dicker[3,4], Caroline J. Jagtenberg[5]

1 Rotterdam School of Management, Erasmus University Rotterdam, Rotterdam, The Netherlands, 2 School of Operations Research and Information Engineering, Cornell University, Ithaca, New York, United States of America, 3 Clinical Audit and Research, Hato Hone St John New Zealand, Auckland, New Zealand, 4 Paramedicine Research Unit, Paramedicine Department, Auckland University of Technology, Auckland, New Zealand, 5 School of Business and Economics, Vrije Universiteit, Amsterdam, The Netherlands

* vandenberg@rsm.nl

## Abstract

### Background

Community First Responders (CFRs) are commonly used for out-of-hospital cardiac arrests, and advanced systems send so-called phased alerts: notifications with built-in time delays. The policy that defines these delays affects both response times and volunteer fatigue.

### Methods

We compare alert policies by Monte Carlo Simulation, estimating patient survival, coverage, number of alerts and redundant CFR arrivals. In the simulation, acceptance probabilities and response delays are bootstrapped from 29,307 rows of historical data covering all GoodSAM alerts in New Zealand between 1-12-2017 and 30-11-2020. We simulate distances between the patient and CFRs by assuming that CFRs are located uniformly at random in a 1-km circle around the patient, for different CFR densities. Our simulated CFRs travel with a distance-dependent speed that was estimated by linear regression on observed speeds among those responders in the above-mentioned data set that eventually reached the patient.

### Results

The alerting policy has a large impact on the four metrics above, and the best choice depends on volunteer density. For each volunteer density, we are able to identify a policy that improves GoodSAM New Zealand's current policy on all four metrics. For example, when there are 30 volunteers within 1 km from the patient, sending out alerts to 7 volunteers and replacing each volunteer that rejects by a new one, is expected to save 10 additional lives per year compared to the current policy, without

**Data availability statement:** The data that support the findings of this study are available from Hato Hone St John but restrictions apply to the availability of these data, which were used under license for the current study, and so are not publicly available. Data can however be requested from Hato Hone St John upon reasonable request. Requests can be send to dataandresearchrequests@stjohn.org.nz. More information can be found here: https://www.stjohn.org.nz/news--info/our-performance/clinical-audit-and-research/research-stjohn/.

**Funding:** This study was financially supported by the Dutch Institute for Advanced Logistics (TKI Dinalog) in the form of a grant (2023-1-307TKI) received by CJJ. This study was also financially supported by the Netherlands Organization for Scientific Research (NWO) in the form of grants (VI.Veni.191E.005 and VI.Vidi.241E.016) received by PLvdB. This study was also financially supported by the (USA) National Science Foundation in the form of grants (CMMI-2035086 and OAC 2410950) received by SGH.

**Competing interests:** The authors have declared that no competing interests exist.

increasing volunteer fatigue. Our results also shed light on polices that would improve one metric while worsening another, for example, when there are 10 volunteers within 1 km from the patient, dispatching them all immediately increases our survival estimate by 11% compared to the current policy, with the downside of also increasing the redundant arrivals by 137%.

## Conclusions

Monte Carlo simulation can help CFR system managers identify a good policy before implementing it in practice. We recommend balancing survival and volunteer fatigue, aiming to ultimately further improve a CFR system's effectiveness.

## 1. Introduction

Survival for out-of-hospital cardiac arrest (OHCA) can be significantly improved through bystander efforts [1]. To shorten the time to good-quality cardiopulmonary resuscitation (CPR), some emergency call centers use mobile phone technology to rapidly locate and alert nearby trained volunteers. A number of such community first responder (CFR) systems are active in, for example, the United States [2], the United Kingdom [3] and Europe [4].

An alerting policy prescribes which volunteer to alert when. Simple alerting policies send all alerts at once. However, one may also use *phased* alerts, sending them in batches with time lags in between, to see if previous ones have been accepted. Such a policy is used by Hato Hone St John New Zealand, an ambulance provider that deploys the CFR system GoodSAM. Hato Hone St John New Zealand configured the system to dispatch CFRs in batches of 3 with time lags of 1 minute.

### 1.1. Importance

While for an individual patient, alerting many volunteers is beneficial, this leads to a high alert rate for volunteers as well as an increased likelihood of multiple volunteers arriving on scene, which may diminish their perceived contribution. Both aspects can lead to a long-term negative impact on volunteers' willingness to respond (volunteer fatigue), which would ultimately lower future OHCA survival. It is therefore important to find a good balance between future volunteer fatigue and current volunteer response times.

### 1.2. Goal of this investigation

This paper aims to assist CFR system managers in understanding the consequences of a phased alerting policy. To that end, we define a large set of policies that are not restricted by any particular existing setup, and would for many CFR systems likely require only a small software change. We track five different Key Performance Indicators (KPIs): expected patient survival, coverage, number of alerts, number of redundant arriving volunteers and patients with two or more arriving volunteers. We evaluate all policies through Monte Carlo simulation using historical data from

GoodSAM responses in New Zealand. Finally, we emphasize how the optimal choice depends on volunteer density. This is the first study to use analytics to systematically predict the performance of different alerting policies, thereby estimating the potential of phased alerting, if used well.

## 2. Methods

### 2.1. Setting

The process of both CFR and EMS response is depicted in Fig 1. There, the name `EMS response time' reflects the duration that is common for EMS providers to measure, and for which they often face targets and reporting obligations. For clarity, we have also added a `GoodSAM response time', which starts at the moment the CFR system is activated.

EMS response times are measured from the moment of call arrival, so they are not ideal for estimating OHCA survival: the time that has passed since OHCA onset is more relevant. To that end, we define another duration, $T_{EMS}$: the duration between OHCA onset and EMS arrival. Similarly, we define $T_{CPR}$: the duration between OHCA onset and the arrival of the first responder, regardless of whether this is a volunteer or EMS. This is summarized in Fig 1.

### 2.2. Study design

We defined a number of alerting policies, see Section 2.3. To estimate their performance, we used historical records of GoodSAM responses in New Zealand (Section 2.4) and applied them in a Monte Carlo simulation (Section 2.5) to quantify the GoodSAM response time (see Fig 1). To advance generalizability, we defined three different studies that correspond to different volunteer densities: low (10 volunteers within 1 km), medium (30 volunteers within 1 km) and high (100 volunteers within 1 km). These densities roughly correspond to what would be found in Auckland, New Zealand if 0.2% of the inhabitants signed up (low), what is recommended [5] (medium) and our estimate for one of the most volunteer-dense regions worldwide (high). For each policy and each density, we use the GoodSAM response time obtained from simulation and convert them to expected values of four key performance indicators (KPIs, see Section 2.6).

### 2.3. Policies

We consider so-called alerting policies, which may base the decision to send an alert on the amount of time that has passed, and/or the responses that are received from previously alerted volunteers. Our policies only alert a more distant volunteer when all closer volunteers are already alerted. This helps to narrow down the set of all policies to realistic and

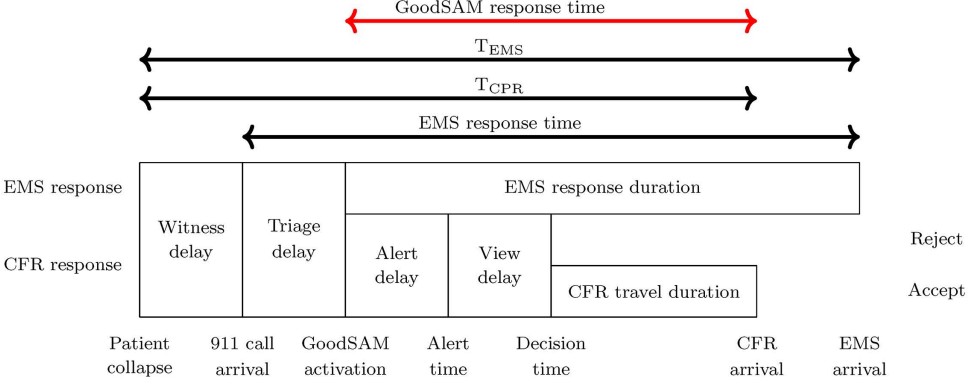

**Fig 1. CFR response process.**

promising ones. We further assume that no alerts are sent out after a volunteer accepts an alert, or 10 minutes have passed after system activation, whichever comes first.

We evaluate the following policies:

- *Send all at time 0*: alert all available volunteers within the dispatch radius immediately upon system activation, which we call time 0.

- *Send n1 alerts at time 0*: Alert $n1$ volunteers immediately at time 0, and send no alerts after.

- *Keep n2 alerts active*: Alert $n2$ volunteers at time 0 and replace every incoming reject with an additional alert.

- *NZ current policy*: Alert 3 volunteers at time 0 and 3 additional volunteers every minute until one volunteer accepts the alert, or 10 minutes have passed.

For *send n1 alerts at time 0*, we consider the values of $n1$ between 1 and 15. For *Keep n2 alerts active* we consider values of $n2$ between 1 and 10. This gives a total of 27 policies.

## 2.4. Data

We used data obtained from the Hato Hone St John Ambulance Service, which operates the GoodSAM app in New Zealand. For more information on the system, see S1 Table in the supporting information. The data covers all OHCAs in the country between 1-12-2017 and 30-11-2020, along with their location and 911 call arrival time and was accessed on 15-01-2021. The data also contains which CFRs were alerted through GoodSAM, at what time, and what their location was at the time of notification (*alert time* in Fig 1). Moreover, the data contains the time at which the volunteer reacted (*decision time*) by either accepting or rejecting the alert. Finally, the time volunteers arrived on scene (*CFR arrival*) is determined by the system, based on GPS signal. A GPS location within 50 meters of the scene is recorded as on-scene. Responders can also manually press a button to generate an on-scene timestamp. The data contains a total of 29,307 CFR alerts, which were either accepted (4009), accepted and dropped later (1199), rejected (7925), or not seen (16,174). From the 4009 accepting volunteers, 1776 eventually reached the patient. The authors did not have access to information that could identify individual patients or volunteers.

For the view delay, we measured the duration between the alert time and the decision time. We ignored entries for which the alert time was equal to 'nan' (this excludes 1 entry). If the decision time was not listed, which is the case for not-seen alerts, we set the view delay to infinity. We treated both accepted and accepted-but-dropped-later entries as accepts, and we treated rejected and not-seen as rejects. This led to an empirical acceptance rate of $\frac{4009+1199}{29,307} \approx 17.77\%$. Fig 2 shows the empirical view delay distribution, as well as the estimated empirical probability of an accept, given a certain view delay. Not-seen alerts are not visible in this plot as these have an infinite view delay.

The CFR's travel duration depends on their distance from the patient. Locations and travel times were only recorded for *alerted* CFRs, so we cannot observe information on non-alerted individuals. However, in our counterfactual simulations those CFRs *may* be alerted, so we estimate their distances using Monte Carlo simulation. For each CFR that is within 1 km of the patient, we simulate their location uniformly at random in this circle. This is also at the heart of a so-called Poisson process, which was previously proposed as a suitable model for CFR locations [6] and used to represent the randomness of OHCA locations over time [7].

To translate distance into travel time, we used empirical data on volunteer travel speed, conditioning on the distance between volunteer and patient. We excluded 14 observations with a travel time of 0 seconds. We discretized the distances in 100-meter blocks and for each block, took the median speed of all volunteer responses in the data. We excluded the data for responses between 0 and 100 meters as we considered these less reliable. We then performed linear regression on the remaining medians to estimate the relationship between distance and speed.

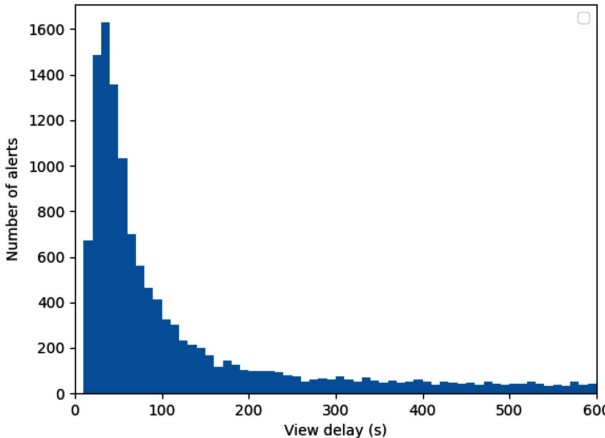 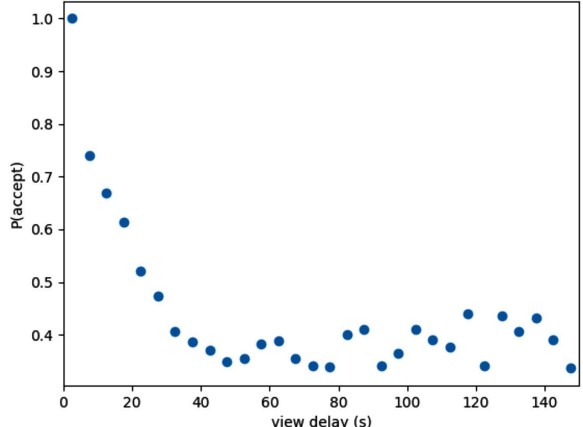

**Fig 2. Historical view delays and acceptance probabilities as observed on GoodSAM in New Zealand.**

## 2.5. Simulation

Our Monte Carlo simulation considers two sources of uncertainty: 1) the distance between the volunteers and the patient, and 2) the view delay and reply of alerted volunteers. The simulation generates 1,000 different OHCAs. For each OHCA, the simulation draws $n$ random volunteer locations (for $n = 10, 30, 100$) uniformly distributed across the 1-km circle. The straight-line distance between these locations and the patient are translated to travel time using the regression described in Section 2.4, resulting in the relationship as described in Equation (1).

The timings of the alerts are simulated according to their policy definitions (Section 2.3). For each volunteer that receives an alert, the view delay and corresponding reply (accept/reject/not seen) was bootstrapped 10,000 times from the empirical data. The view delay and reply were jointly sampled to maintain the statistical dependence between the time a volunteer takes to respond and the likelihood that the reply is `accept`, see Fig 2.

Each *accepting* volunteer's alert time plus view delay plus their travel time gives a volunteer response time. For each patient, we only store the arrival time of the first-arriving volunteer. The time until ambulance arrival, $T_{EMS}$, is assumed to be 13 minutes (this is Hato Hone St John New Zealand's target for urgent requests [8] as well as the median observed EMS time for rural areas in 2020/2021 [9]).

We estimate the time to CPR ($T_{CPR}$, see Fig 1) as follows. We add a 1-minute `witness delay` and a 124-sec `triage delay` (the median of the data) to the simulated response time of the first-arriving volunteer, truncated by $T_{EMS}$. This is converted to a survival probability according to the regression model

$$survival\ (T_{CPR},\ T_{EMS}) = \ 1 - \left(1 + exp\ \left\{0.04 + 0.3\ T_{CPR} + 0.14\ (T_{EMS} - T_{CPR})\right\}\right)^{-1}$$

that was derived by Waalewijn et al [10]. The resulting survival probability was converted to lives saved by multiplying it by 5,141, the nationally observed number of OHCAs per year in New Zealand [11].

## 2.6. Outcomes

We consider two KPIs related to the quality of care and two KPIs related to the inconvenience caused to volunteers.

**Coverage:** fraction of incidents that receive a CFR response within a given time threshold. We define this on the `GoodSAM response time` (see Fig 1) and use a threshold of 5 minutes.

**Lives saved:** the number of OHCA patients that are expected to survive until hospital discharge, in New Zealand, annually.

**Number of alerts:** the average number of alerts sent per incident.

**Redundant arrivals:** the average number of volunteers arriving on scene after the first responder, per incident.

**Patients with 2+ arrivals:** the fraction of patients where two or more volunteers arrive.

## 3. Results

### 3.1. Parameter estimation

Fig 3a shows the observed median travel speeds for different distances, which increase with distance. Linear regression of travel speeds in km/h, $y$, on distances in meters, $x$, based only on the distances above 100 meters has an $R^2$ of 0.96 and yields the relationship

$$y = 1.83 + 0.0108x.$$

The resulting relationship between the distance in meters and travel time in minutes, plotted in Fig 3b, is

$$travelTime(x) = \frac{60x}{1830 + 10.8x}. \tag{1}$$

### 3.2. Performance of policies

Tables 1–3 show the results for the different policies for a low, medium and high number of volunteers. For reference, alerting 0 volunteers would lead to 98 survivors, 0 redundant arrivals, 0 alerts and 0 coverage.

By only alerting the closest volunteer, the number of survivors can be increased to 120, 124, or 129, for 10, 30 or 100 volunteers within 1 km, respectively. For coverage, the contribution of the closest volunteer is 11.8%, 13.4%, or 14.5%, respectively.

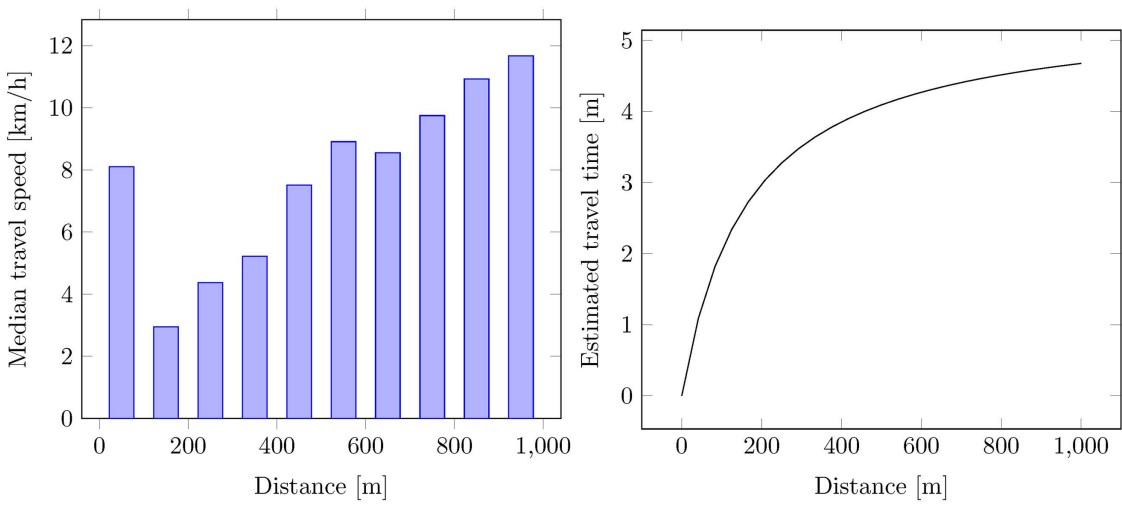

(a) Median travel speed depending on distance to patient.

(b) Estimated travel time in minutes depending on distance to patient.

**Fig 3. Distance-dependent travel times estimated based on historical data.**

**Table 1. Results for n = 10 volunteers within 1 km, based on 1000 simulated volunteer locations, with 10000 view delay and acceptance simulations each. The maximum confidence interval halfwidth as a fraction of the mean among all these numbers was 0.012 (ignoring those cases where the mean was zero). Policies that send more than 10 alerts at time 0 are omitted, as there are only 10 eligible volunteers in this situation.**

| Policy | Coverage | Survivors per year | Nr of alerts | Redundant arrivals | 2+arrivals |
|---|---|---|---|---|---|
| Send all at time 0 | 0.420 | 191 | 10 | 0.915 | 0.552 |
| Keep 1 alert active. | 0.124 | 123 | 1.3 | 0 | 0 |
| Keep 2 alerts active. | 0.207 | 140 | 2.6 | 0.041 | 0.041 |
| Keep 3 alerts active. | 0.268 | 153 | 3.8 | 0.114 | 0.107 |
| Keep 4 alerts active. | 0.313 | 163 | 4.9 | 0.21 | 0.185 |
| Keep 5 alerts active. | 0.347 | 171 | 6 | 0.324 | 0.267 |
| Keep 6 alerts active. | 0.373 | 178 | 7.1 | 0.451 | 0.348 |
| Keep 7 alerts active. | 0.391 | 183 | 8 | 0.585 | 0.422 |
| Keep 8 alerts active. | 0.405 | 187 | 8.8 | 0.715 | 0.483 |
| Keep 9 alerts active. | 0.414 | 190 | 9.5 | 0.831 | 0.528 |
| Keep 10 alerts active. | 0.420 | 191 | 10 | 0.915 | 0.552 |
| Send 1 alerts at time 0. | 0.118 | 120 | 1 | 0 | 0 |
| Send 2 alerts at time 0. | 0.202 | 135 | 2 | 0.031 | 0.031 |
| Send 3 alerts at time 0. | 0.265 | 147 | 3 | 0.089 | 0.083 |
| Send 4 alerts at time 0. | 0.312 | 157 | 4 | 0.167 | 0.147 |
| Send 5 alerts at time 0. | 0.347 | 165 | 5 | 0.264 | 0.217 |
| Send 6 alerts at time 0. | 0.373 | 172 | 6 | 0.374 | 0.289 |
| Send 7 alerts at time 0. | 0.391 | 178 | 7 | 0.497 | 0.36 |
| Send 8 alerts at time 0. | 0.405 | 183 | 8 | 0.628 | 0.429 |
| Send 9 alerts at time 0. | 0.414 | 187 | 9 | 0.769 | 0.493 |
| Send 10 alerts at time 0. | 0.420 | 191 | 10 | 0.915 | 0.552 |
| NZ current strategy. | 0.266 | 172 | 7 | 0.384 | 0.313 |

Comparing policies *Send n1 alerts at time 0,* with *Keep n2 alerts active* indicates that a similar level of survival can be obtained with 1 alert fewer at time 0, by replacing rejects with additional alerts. This reduces the impact of alerts on volunteers (both in terms of redundant arrivals as well as the number of alerts) without sacrificing survival.

The current policy used by GoodSAM in New Zealand on average leads to sending out 9.0 alerts per incident, if sufficiently many volunteers are within 1 km from the patient. The corresponding number of expected survivors per year is 172, 189, or 212 for a low, medium and high number of volunteers. Coverage as a result of this policy is estimated at 26.6%, 35.2%, or 50.5%, respectively.

The policy that sends an alert to all volunteers at time 0 by definition yields the highest survival (estimated using the survival function of Waalewijn et al. [10] to be 191, 231 and 259 survivors per year) and coverage (42.0%, 80.7% and 99.6%), but also gives the highest estimated number of redundant arrivals (0.915, 4.324 and 16.737) and alerts (10, 30, 100).

## 4. Discussion

The trade-offs between the KPIs in Tables 1–3 help a CFR system manager identify a desirable dispatch policy. This choice will likely depend on the volunteer density. Our reported number of volunteers within 1 km of the patient has a direct relationship with the volunteer density. The three cases we considered correspond to a density of 3.18 vol/km$^2$, 9.55 vol/km$^2$, and 31.83 vol/km$^2$, respectively. For reference, the literature recommends a volunteer density of at least 10 vol/km$^2$ [5].

**Table 2. Results for n = 30 volunteers within 1 km, based on 1000 simulated volunteer locations, with 10000 view delay and acceptance simulations each. The maximum confidence interval halfwidth as a fraction of the mean among all these numbers was 0.009 (ignoring those cases where the mean was zero).**

| Policy | Coverage | Survivors per year | Nr of alerts | Redundant arrivals | 2+ arrivals |
|---|---|---|---|---|---|
| Send all at time 0 | 0.807 | 231 | 30 | 4.324 | 0.979 |
| Keep 1 alert active. | 0.149 | 128 | 1.3 | 0 | 0 |
| Keep 2 alerts active. | 0.258 | 148 | 2.6 | 0.041 | 0.041 |
| Keep 3 alerts active. | 0.344 | 164 | 3.8 | 0.114 | 0.107 |
| Keep 4 alerts active. | 0.413 | 176 | 4.9 | 0.21 | 0.185 |
| Keep 5 alerts active. | 0.47 | 185 | 6 | 0.325 | 0.267 |
| Keep 6 alerts active. | 0.518 | 193 | 7.1 | 0.454 | 0.35 |
| Keep 7 alerts active. | 0.558 | 200 | 8.2 | 0.593 | 0.428 |
| Keep 8 alerts active. | 0.593 | 205 | 9.2 | 0.74 | 0.501 |
| Keep 9 alerts active. | 0.622 | 209 | 10.2 | 0.894 | 0.568 |
| Keep 10 alerts active. | 0.648 | 213 | 11.2 | 1.052 | 0.628 |
| Send 1 alerts at time 0. | 0.134 | 124 | 1 | 0 | 0 |
| Send 2 alerts at time 0. | 0.241 | 142 | 2 | 0.031 | 0.031 |
| Send 3 alerts at time 0. | 0.327 | 157 | 3 | 0.089 | 0.083 |
| Send 4 alerts at time 0. | 0.399 | 169 | 4 | 0.167 | 0.147 |
| Send 5 alerts at time 0. | 0.459 | 178 | 5 | 0.264 | 0.217 |
| Send 6 alerts at time 0. | 0.509 | 186 | 6 | 0.374 | 0.289 |
| Send 7 alerts at time 0. | 0.551 | 193 | 7 | 0.497 | 0.36 |
| Send 8 alerts at time 0. | 0.588 | 199 | 8 | 0.629 | 0.429 |
| Send 9 alerts at time 0. | 0.619 | 204 | 9 | 0.769 | 0.493 |
| Send 10 alerts at time 0. | 0.645 | 208 | 10 | 0.915 | 0.552 |
| Send 11 alerts at time 0. | 0.668 | 211 | 11 | 1.068 | 0.606 |
| Send 12 alerts at time 0. | 0.688 | 214 | 12 | 1.224 | 0.656 |
| Send 13 alerts at time 0. | 0.705 | 216 | 13 | 1.385 | 0.7 |
| Send 14 alerts at time 0. | 0.72 | 219 | 14 | 1.548 | 0.739 |
| Send 15 alerts at time 0. | 0.733 | 221 | 15 | 1.714 | 0.774 |
| NZ current strategy. | 0.352 | 189 | 9 | 0.582 | 0.441 |

Because a different density might lead to a different preferred policy, the GoodSAM system offers the ability to configure different rules for urban and rural areas. This is done by uploading KML files that represent parts of the map. This feature implies that potential insights from this paper can readily be implemented in practice.

Alerting the same number of volunteers leads to higher survival estimates and better coverage as volunteer density increases. This effect is visible throughout Tables 1–3, and also holds in generality, assuming all other system parameters remain equal. The reason is that the $n$-th closest volunteer is expected to be nearer to the patient when volunteer density is higher.

Comparing policies *Send 6 alerts at time 0* and *Send 10 alerts at time 0* shows different survival numbers for different volunteer densities. In Table 1 the difference is 19 lives, while in Table 3 it is 26 lives. The reason is that in Table 1, the 6th-10th volunteers are farther from the patient than they are in Table 3.

Perhaps more surprising is that the policy *Send 6 alerts at time 0* gives approximately the same survival estimate as *Keep 5 alerts active*, and that this is the case in each of the three tables. We have no intuition for this, but conjecture that this result would be found for any volunteer density within the entire range of 3.18 -- 31.83 vol/km$^2$.

 

**Table 3. Results for n = 100 volunteers within 1 km, based on 1000 simulated volunteer locations, with 10000 view delay and acceptance simulations each. The maximum confidence interval halfwidth as a fraction of the mean among all these numbers was 0.006 (ignoring those cases where the mean was zero).**

| Policy | Coverage | Survivors per year | Nr of alerts | Redundant arrivals | 2+ arrivals |
|---|---|---|---|---|---|
| Send all at time 0 | 0.996 | 259 | 100 | 16.737 | 1 |
| Keep 1 alert active. | 0.168 | 134 | 1.3 | 0 | 0 |
| Keep 2 alerts active. | 0.297 | 159 | 2.6 | 0.041 | 0.041 |
| Keep 3 alerts active. | 0.399 | 178 | 3.8 | 0.114 | 0.107 |
| Keep 4 alerts active. | 0.483 | 193 | 4.9 | 0.21 | 0.185 |
| Keep 5 alerts active. | 0.552 | 205 | 6 | 0.325 | 0.268 |
| Keep 6 alerts active. | 0.609 | 214 | 7.1 | 0.454 | 0.35 |
| Keep 7 alerts active. | 0.657 | 222 | 8.2 | 0.594 | 0.429 |
| Keep 8 alerts active. | 0.697 | 228 | 9.2 | 0.741 | 0.502 |
| Keep 9 alerts active. | 0.732 | 233 | 10.2 | 0.894 | 0.568 |
| Keep 10 alerts active. | 0.762 | 237 | 11.2 | 1.053 | 0.628 |
| Send 1 alerts at time 0. | 0.145 | 129 | 1 | 0 | 0 |
| Send 2 alerts at time 0. | 0.263 | 151 | 2 | 0.031 | 0.031 |
| Send 3 alerts at time 0. | 0.362 | 169 | 3 | 0.089 | 0.083 |
| Send 4 alerts at time 0. | 0.445 | 184 | 4 | 0.168 | 0.147 |
| Send 5 alerts at time 0. | 0.515 | 196 | 5 | 0.264 | 0.217 |
| Send 6 alerts at time 0. | 0.574 | 205 | 6 | 0.374 | 0.289 |
| Send 7 alerts at time 0. | 0.625 | 213 | 7 | 0.497 | 0.361 |
| Send 8 alerts at time 0. | 0.669 | 220 | 8 | 0.629 | 0.429 |
| Send 9 alerts at time 0. | 0.707 | 226 | 9 | 0.769 | 0.493 |
| Send 10 alerts at time 0. | 0.74 | 231 | 10 | 0.916 | 0.552 |
| Send 11 alerts at time 0. | 0.768 | 235 | 11 | 1.068 | 0.606 |
| Send 12 alerts at time 0. | 0.793 | 238 | 12 | 1.225 | 0.656 |
| Send 13 alerts at time 0. | 0.815 | 241 | 13 | 1.385 | 0.7 |
| Send 14 alerts at time 0. | 0.834 | 244 | 14 | 1.549 | 0.739 |
| Send 15 alerts at time 0. | 0.851 | 246 | 15 | 1.715 | 0.774 |
| NZ current strategy. | 0.505 | 212 | 9 | 0.585 | 0.444 |

Tables 1–3 show that Hato Hone St John's current alerting policy offers room for improvement. For example, with 10 volunteers in the circle (Table 1), this policy is expected to yield 172 survivors and sends, on average, 7 alerts per patient. In Table 1 we identify that the policy *Send 6 alerts at time 0* achieves the same level of survival with a higher coverage and a lower number of alerts and redundant arrivals. Moreover, Tables 2 and 3 show that switching from Hato Hone St John's current strategy to *Send 7 alerts at time 0* is expected to yield a few additional survivors while simultaneously improving the other three KPIs. This insight might be a reason to fine-tune settings in the GoodSAM system.

To consider the generalizability of these results to other countries, we emphasize which parts of this research were New Zealand-specific: the view delay, travel speed and acceptance probability. Acceptance rates in New Zealand were found to be similar to those reported for other CFR smartphone apps [12]. CFR travel speeds and view delays are not readily available for other countries in the literature, but the framework designed for this study allows for easy testing on other parameters. Should readers be interested in whether the same conclusions hold for their country or region, the authors encourage them to reach out and offer to run simulations with different data.

## 4.1. Limitations

A limitation of this study is that we estimate survival through a so-called survival curve that describes a relationship between time to CPR and patient survival. However, this approach implicitly assumes that volunteers provide the same quality CPR as those in the study that estimated the survival curve. Up to today, we are unfortunately still lacking sufficient evidence for the effectiveness of OHCA first responder systems, because while existing systems showed improved rates of bystander CPR, no effect on survival could be proven. For more information see [13].

In this study the time on-scene is based on the geolocation of the phone, which may have some error due to gaps in cell tower coverage. It is unknown how often this situation occurs. The stop-gap to this is that responders can also manually generate an on-scene time by pressing a button in the app, but responders do not always do this.

Another limitation is that we focused on volunteers who directly go to the patient to provide CPR. However, in practice, certain CFR systems differentiate alerts, directing some volunteers to immediately attend to the patient and others to retrieve an Automated External Defibrillator (AED). Such a system raises new questions; answering those questions would require an extension of our methods.

We explored results for three different volunteer densities, but a CFR system will have varying volunteer densities throughout the day and the region. It might then be worthwhile to differentiate the dispatch policy within one system through times of the day and parts of the region. Accurately evaluating such a CFR system would require running simulations with the correct mix of densities and policies, thereby introducing complexity and likely diminishing insight. Therefore we decided not to pursue that approach here.

## 4.2. Future research

Although it is plausible that a high number of alerts and/or redundant arrivals reduce future CFR engagement, there is, to the best of our knowledge, no study that quantifies the magnitude of this effect. We welcome such studies, as they might aid in designing even better dispatch policies. The current knowledge gap led us to refrain from recommending a specific dispatch policy, instead presenting a list of options from which a CFR system manager can choose.

Future research might address more sophisticated policies. For example, consider a CFR as inactive when they have not responded for some time, or let the policy depend on the real-time locations of volunteers. It would also be interesting to discover the value of incorporating volunteer-specific knowledge, such as their historical acceptance rate, in the decision of which volunteer to dispatch. One may also seek improvement by, for example, asking volunteers for their travel mode as is done in Germany [14]. Finally, one might design a separate night-time policy that is trained on lower acceptance rates.

## 5. Conclusions

A CFR system's dispatch policy affects the trade-off between the number of alerts, redundant arrivals, and time to CPR. In comparison to sending all alerts immediately, it is beneficial to send a reduced number of alerts immediately plus additional alerts upon receiving rejections. By doing this, a policy can be constructed to improve Hato Hone St John's current policy on *all* four KPIs (sending out alerts to 7 volunteers and replacing each volunteer that rejects by a new one, is expected to save 11 additional lives per year compared to the current policy, without increasing volunteer fatigue, see Table 2).

When quantifying KPIs and drawing conclusions, it is crucial to account for the volunteer density. For example, the conclusion above based on Table 2 holds when there are 30 volunteers within 1 km from the patient. CFR system managers should familiarize themselves with volunteer density in their region, before identifying a dispatch policy that suits them.

Besides volunteer density, the ideal dispatch policy depends on how one wishes to trade off KPIs. Some policies would improve one metric while worsening another, for example, when there are 10 volunteers within 1 km from the patient, dispatching them all immediately increases our survival estimates by $(191-172)/172 = 11\%$ compared to the current policy, with the downside of also increasing the redundant arrivals by $(0.918-0.384)/0.384 = 137\%$.

As volunteer densities may vary over the day and between different parts of the region, the choice may even be differentiated within one system. The GoodSAM system currently already allows for such user-specified geographical policy differentiation.

We hope that this article will stimulate discussion on phased dispatching of CFRs and pave the way for incorporating methods from the field of operations research in designing improved dispatch policies.

## Supporting information

**S1 Table. Information on the GoodSAM New Zealand first responder system.** Information reported in alignment with recommended reporting items for Community First Responder (CFR) systems (10.1016/j.resuscitation.2023.110087). (PDF)

## Acknowledgments

We thank GoodSAM and Hato Hone St John Ambulance Service New Zealand for access to data.

## Author contributions

**Conceptualization:** Pieter L. van den Berg, Shane G. Henderson, Hemeng Li, Bridget Dicker, Caroline J. Jagtenberg.

**Data curation:** Bridget Dicker.

**Methodology:** Pieter L. van den Berg, Shane G. Henderson, Hemeng Li, Caroline J. Jagtenberg.

**Software:** Pieter L. van den Berg, Hemeng Li, Caroline J. Jagtenberg.

**Writing – original draft:** Pieter L. van den Berg, Caroline J. Jagtenberg.

**Writing – review & editing:** Shane G. Henderson, Hemeng Li, Bridget Dicker.

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
