## [Decision Letter · Decision Letter 0]

29 Apr 2025

Dear Dr. van den Berg,

Thank you for submitting your manuscript to PLOS ONE. After careful consideration, we feel that it has merit but does not fully meet PLOS ONE’s publication criteria as it currently stands. Therefore, we invite you to submit a revised version of the manuscript that addresses the points raised during the review process.

Based on the reviewers' feedback, significant revisions are required before further consideration. Please address all comments thoroughly and submit a point-by-point response along with a revised manuscript.

We look forward to receiving your revised manuscript.

Kind regards,

Ernesto Iadanza

Academic Editor

PLOS ONE

Journal Requirements:

[This work was supported by the Dutch Institute for Advanced Logistics (TKI Dinalog) [grant number 2023-1-307TKI], the Netherlands Organization for Scientific Research (NWO) [grant number VI.Veni.191E.005] and the (USA) National Science Foundation [grant number CMMI-2035086].].

4. In the online submission form, you indicated that [The data that support the findings of this study are available from Hato Hone St John but restrictions apply to the availability of these data, which were used under license for the current study, and so are not publicly available. Data are however available from the authors upon reasonable request and with permission of Hato Hone St John.].

Reviewers' comments:

Reviewer's Responses to Questions

**Comments to the Author**

1. Is the manuscript technically sound, and do the data support the conclusions?

Reviewer #1: Yes

Reviewer #2: No

2. Has the statistical analysis been performed appropriately and rigorously?

Reviewer #1: N/A

Reviewer #2: Yes

3. Have the authors made all data underlying the findings in their manuscript fully available?

Reviewer #1: No

Reviewer #2: No

4. Is the manuscript presented in an intelligible fashion and written in standard English?

Reviewer #1: Yes

Reviewer #2: Yes

Reviewer #1: The authors present the results of a simulation-based study that investigates the impact of different phased alerting strategies on Community First Responder (CFR) systems.

This study offers an interesting approach to addressing this relevant issue, and the findings could potentially influence CFR alerting strategies, albeit within the outlined limitations.

Although the simulations are complex and challenging to reproduce, the detailed explanation of methods, including the parameters and metrics, ensures that the derivation of the results as well as the conclusions are clear and understandable.

A key question regarding the relevance of this study is the uncertainty regarding the impact of unnecessary alarms on the motivation of first responders. While an increase in the number of registered responders may lead to more redundant arrivals depending on the alerting strategy, it also allows these arrivals to be distributed across more individuals, potentially reducing the negative effect on motivation. In regions with a relatively low density of first responders, however, the findings presented in the manuscript become even more significant.

In their submission, the authors provided a comprehensive explanation of why the data on which the study is based is not publicly available and that it can be viewed on request.

Some minor adaptions are recommended:

Give a definition of the abbreviation KPI the first time it is used.

I would recommend omitting the calculation of the ratios in lines 36 and 37 of the abstract and simply presenting the results as 11% and 137%.

Provide a y-axis label in the left panel of Figure 2.

Consider creating a comprehensive figure that graphically illustrates the processes and scenarios analyzed. This could help make the results of the simulations more tangible for the reader.

Reviewer #2: Thank you for the opportunity to review the manuscript entitled “Phased alerting of community first responders for cardiac arrest“ by van den Berg and colleagues. The authors aimed to compare different alert policies in the first responder system GoodSam via simulation. The authors based their assumptions for the simulation partly on historical data from real GoodSam alerts in New Zealand. The data extraction was performed on January 15th, 2021. The manuscript explores a topic that should be of interest to the readers of PLOS ONE.

Unfortunately, I have to raise several major concerns regarding the manuscript. Multiple statements are overstated and do not follow current reporting standards for first responder systems. Furthermore, the transferability of the results is highly doubtful.

Major comments:

Introduction

1. Lines 51-56: It is difficult to describe the simple use of phased alerts as “complex alerting”. Several systems today already take into account the expected arrival time of the emergency services and the current traffic situation, as well as the means of transport of the first responder. The reader should be made aware that more complex alert algorithms are already possible. The study and the results refer to the GoodSam system only. The transferability of the results to other systems is extremely limited.

2. Lines 58-63: Volunteer fatigue can theoretically pose a problem in a first responder system. However, there is no study that has ever shown that volunteer fatigue occurs or that there are limits to, for example, false alarms or arriving at the scene with other responders, which trigger volunteer fatique. Or could the authors find any evidence of volunteer fatique in their own historical data? If so, how did this affect the historical data on which your model is based?

3. In my personal opinion, the chance to trigger volunteer fatique is a lot greater, if volunteers arrive after EMS than multiple first responders arriving before EMS on scene. How could this be implemented in your simulation?

4. This is not the first study to evaluate different alerting policies in a community first responder system (also refer to: 10.1007/s10049-024-01395-2). This might be the first study to evaluate alerting policies in GoodSam. Please clarify.

5. Please refer to the reporting standard for describing first responder systems, smartphone alerting systems, and AED networks (10.1016/j.resuscitation.2023.110087). Several core parameters are missing in your manuscript and need to be included to get an overview of the first responder system in New Zealand.

Methods

6. Lines 102-103: Is it correct, that given “that no alerts are sent out after a volunteer accepts an alert”. Most cases would only have one first responder on scene? To perform high quality chest compressions, current guidelines recommend changing ever two minutes. Therefore, at least two responders on scene are needed. The rate of alarms with at least 2 responders on scene should be included as an KPI in your model.

7. Lines 102-103: “or 10 minutes have passed after GoodSAM activation”. Since the expected time of EMS arrival seems to be unavailable in GoodSam and an EMS arrival time of 13 minutes after the initial alarm is assumed (compare line 166), it is surprising that the authors chose 10 minutes as a cut-off to stop alerting. Taking into account the authors’ historical data (Figure 3b), a first responder alerted after 9 or 10 minutes would need to be within 200 m of the patient to arrive within 3 minutes (before the EMS) and to have an impact on patient survival. Please also refer to comment #3 in this context. Also: Since the authors assume a witness and triage delay of 3 minutes in all cases (Lines 168-170), the EMS would already have arrived, when alerting a first responder after 10 minutes.

8. Line 122-123: Please clarify what GPS radius was used to measure the CFR's arrival at the scene. The chosen GPS radius has a significant impact on the accuracy of your data (10.1080/10903127.2021.1983094).

9. Lines 123-132: Please provide further information on your historical data as recommended by the reporting standard (see comment #5). For example, activation rate, response rate and arrival before EMS rate are missing core parameters.

10. Lines 170-175: The calculated survival probability, which is included as a KPI, is a very poor estimation. Please provide data on the observed survival in the observation period given the current first responder system alert policy. How does this data compare to the calculated survival probability?

11. Lines 173-175 and 166: It is surprising that the authors chose to rely on data from the ministry of health or “EMS targets” rather than the existing OHCA registry report from the time the historical data was extracted (e.g.: https://www.resus.org.nz/assets/OHCA_All_NZ_March_2022_HQ.pdf). I recommend relying on the available real-world data and to redo the analysis. The current data basis for the basic assumptions of the simulation is simply not transferable. Please explain.

Conclusion and Abstract

12. Line 38: Please rewrite the following statement: Monte Carlo simulation is suitable to predict how changing a CFR alert policy affects survival and volunteer fatigue. None has been shown in your manuscript. The transferability of your results is highly doubtful.

13. Lines 30-33: Up to today, we are unfortunately still lacking sufficient evidence to claim that first responder systems improve survival after OHCA. So far only improved rates of bystander CPR could be proven with no effect on survival. Please also refer to the current AHA or ERC guidelines or ILCOR for further information. Your results are therefore completely overstated, especially in light of comment #11.

14. In your discussion: Since you acknowledge that volunteer density may vary regionally and depending on the time of day, how should CFR system managers estimate the volunteer density in advance? Please explain. In my personal opinion, a “smart” alerting system needs to check the CFR density within its alerting algorithm.

Minor comments:

15. Please include “GoodSam“ and “Simulation“ in your title to clarify the study content.

16. Lines 113/114: Please use n1 and n2 to clarify which n can have what value in lines 107-112.

17. Line 196: x is the distance in meters? Please clarify.

18. Figure 3b: y axis is missing units. Please clarify.

**Do you want your identity to be public for this peer review?** For information about this choice, including consent withdrawal, please see our Privacy Policy

Reviewer #1: No

Reviewer #2: No

---

## [Author Response · Author response to Decision Letter 1]

12 Jul 2025

See attached file for a point-by-point response to the comments raised by the reviewers.

---

## [Decision Letter · Decision Letter 1]

6 Jan 2026

Dear Dr. van den Berg,

Thank you for submitting your manuscript to PLOS ONE. After careful consideration, we feel that it has merit but does not fully meet PLOS ONE’s publication criteria as it currently stands. Therefore, we invite you to submit a revised version of the manuscript that addresses the points raised during the review process.

Based on the reviewers’ comments and the editorial assessment, I am pleased to inform you that your manuscript is **accepted subject to minor revisions** . The required changes are limited to improvements in **formatting consistency, sentence structure, and overall sentence flow**plosone@plos.org . A letter that responds to each point raised by the academic editor and reviewer(s). You should upload this letter as a separate file labeled 'Response to Reviewers'.A marked-up copy of your manuscript that highlights changes made to the original version. You should upload this as a separate file labeled 'Revised Manuscript with Track Changes'.An unmarked version of your revised paper without tracked changes. You should upload this as a separate file labeled 'Manuscript'.

We look forward to receiving your revised manuscript.

Kind regards,

Hari Murthy, Ph.D.

Academic Editor

PLOS One

Journal Requirements:

Additional Editor Comments:

Dear Authors,

Thank you for submitting your manuscript to the journal. The paper has now been evaluated through the editorial review process.

Based on the reviewers’ comments and the editorial assessment, I am pleased to inform you that your manuscript is **accepted subject to minor revisions**. The required changes are limited to improvements in formatting consistency, sentence structure, and overall sentence flow to enhance clarity and readability. No additional experiments or substantive changes to the technical content are required.

Please carefully address the editorial and reviewer suggestions and submit a revised final version of the manuscript within 7 days from the date of this communication. When submitting the revised manuscript, ensure that all formatting adheres strictly to the journal’s guidelines and that the language revisions improve coherence without altering the scientific meaning.

Once the revised version is received and verified, the manuscript will be processed for final acceptance.

We appreciate your interest in publishing with the journal and look forward to receiving your revised manuscript.

Reviewers' comments:

Reviewer's Responses to Questions

**Comments to the Author**

Reviewer #1: All comments have been addressed

2. Is the manuscript technically sound, and do the data support the conclusions?

Reviewer #1: Yes

3. Has the statistical analysis been performed appropriately and rigorously?

Reviewer #1: N/A

4. Have the authors made all data underlying the findings in their manuscript fully available?

Reviewer #1: No

5. Is the manuscript presented in an intelligible fashion and written in standard English?

Reviewer #1: Yes

Reviewer #1: I thank the authors for the comprehensive revision of the manuscript. Their responses to the reviewers’ comments are well reasoned, and the provided clarifications appear appropriate and convincing. Addressing a relevant topic, this simply designed study still makes a useful contribution to the field in the rapidly developing Community First Responder community. Within the limitations discussed by the authors, the conclusions are sound and can be supported. The only remaining remark concerns the optional graphical illustration of the study design, which in its current form adds little value and may be omitted. Overall, I consider the manuscript suitable for publication.

**Do you want your identity to be public for this peer review?** For information about this choice, including consent withdrawal, please see our Privacy Policy

Reviewer #1: No

---

## [Author Response · Author response to Decision Letter 2]

9 Jan 2026

See attached file for a point-by-point response to the comments raised by the reviewers.

---

## [Editor Report · Decision Letter 2]

11 Feb 2026

Simulation of phased alerting of community first responders for cardiac arrest

PONE-D-24-38655R2

Dear Dr. van den Berg,

We’re pleased to inform you that your manuscript has been judged scientifically suitable for publication and will be formally accepted for publication once it meets all outstanding technical requirements.

Kind regards,

Hari Murthy, Ph.D.

Academic Editor

PLOS One

---

## [Editor Report · Acceptance letter]

PONE-D-24-38655R2

PLOS One

Dear Dr. van den Berg,

I'm pleased to inform you that your manuscript has been deemed suitable for publication in PLOS One. Congratulations! Your manuscript is now being handed over to our production team.

Kind regards,

on behalf of

Dr. Hari Murthy

Academic Editor

PLOS One